# Pathogenicity and Interspecies Transmission of Cluster 3 Tembusu Virus Strain TMUV HQ-22 Isolated from Geese

**DOI:** 10.3390/v15122449

**Published:** 2023-12-17

**Authors:** Qing Yang, Yingying Ding, Weiping Yao, Shuyue Chen, Yaqian Jiang, Linping Yang, Guangbin Bao, Kang Yang, Shinuo Fan, Qingqing Du, Qing Wang, Guijun Wang

**Affiliations:** 1College of Animal Science and Technology, Anhui Agricultural University, Hefei 230036, China; yangqing2023@stu.ahau.edu.cn (Q.Y.); dingyingying@stu.ahau.edu.cn (Y.D.); yaoweiping@stu.ahau.edu.cn (W.Y.); chenshuyue@stu.ahau.edu.cn (S.C.); jiangjiang0905@stu.ahau.edu.cn (Y.J.); yanglinping@stu.ahau.edu.cn (L.Y.); baoguangbin@stu.ahau.edu.cn (G.B.); yangkang@stu.ahau.edu.cn (K.Y.); fanshinuo@stu.ahau.edu.cn (S.F.); duqingqing@stu.ahau.edu.cn (Q.D.); 2Anhui Province Key Laboratory of Veterinary Pathobiology and Disease Control, Hefei 230036, China

**Keywords:** Tembusu virus, goose, cluster 3, pathogenicity, interspecies transmission

## Abstract

Since 2010, the Tembusu virus (TMUV) has been highly prevalent in China, causing significant economic losses to the poultry industry. In 2022, a suspected outbreak of TMUV occurred at a goose farm located in Anhui Province. A strain of TMUV, TMUV HQ-22, was isolated from the infected geese. Phylogenetic analysis using the E gene of the HQ-22 strain demonstrated its affiliation with cluster 3, a less commonly reported cluster in comparison to the main circulating cluster, cluster 2. Through a comparison of the envelope (E) protein of HQ-22 with other typical TMUV strains, a mutation at the 157th amino acid position was identified, wherein valine (V) in cluster 3 changed to alanine (A), a characteristic that is unique to cluster 2. These findings highlight the diversity and complexity of the TMUV strains circulating in China. In our experimental analysis, an injection of TMUV HQ-22 into the muscles of 3-day-old goslings resulted in severe neurological symptoms and a mortality rate of 60%. Similarly, the intracranial or intranasal infection of 3-week-old ICR mice with TMUV HQ-22 led to severe neurological symptoms and respective mortality rates of 100% or 10%. In summary, our study isolated a TMUV strain, TMUV HQ-22, from geese that belongs to cluster 3 and exhibits significant pathogenicity in both goslings and ICR mice. These results emphasize the genetic diversity of the TMUV circulating in China and expand the host range beyond mosquitoes to include ducks, chickens, geese, and even mice. It is crucial to not underestimate the risk of TMUV infection in mammals, warranting our utmost attention.

## 1. Introduction

Tembusu virus (TMUV) is an emerging mosquito-borne virus in China. It falls under the Flaviviridae family and the Flavivirus genus, alongside Japanese encephalitis virus (JEV), Dengue virus (DENV), and West Nile virus (WNV) [1]. Infection with TMUV in ducklings and adult ducks is characterized by the development of viral encephalitis, manifesting as symptoms such as depression, ataxia, and other neurological issues [2]. Severe cases may result in paralysis and fatal exhaustion. The earliest known isolated strain of TMUV, TMUV MM1775, was derived from mosquitoes in Malaysia in 1955. However, no subsequent outbreaks or associated diseases have been reported. Only sporadic instances of TMUV isolation in Southeast Asian nations were reported over the ensuing decades [3,4]. In 2000, an outbreak occurred at a broiler chicken farm in Malaysia, leading to a contagious disease marked by ataxia and delayed growth in chicks aged 4–6 weeks. A flavivirus was identified in the impacted chickens and named Sitiawan virus [5]. Since 2010, successive outbreaks of acute infectious diseases have been observed in various duck farming regions in China, including Zhejiang, Shanghai, Jiangsu, and Anhui, among others. The main clinical manifestation of the disease is a significant decline in egg production in laying ducks, with hemorrhagic ovarian inflammation serving as the primary pathological feature [6,7,8,9,10]. There have also been multiple outbreaks of TMUV in duck farming areas in Thailand, resulting in considerable economic losses for both China and Southeast Asia’s duck industry [11,12,13,14].

TMUV is classified as a single-stranded positive-sense RNA virus with an approximate genome length of 11kb. The 5′ untranslated region (UTR) contains a Type I cap structure, while the 3′ UTR lacks a poly-A tail. Within the genome, a single open reading frame (ORF) is present, which is responsible for encoding three structural proteins (C, prM, and E) and seven non-structural proteins (NS1, NS2A, NS2B, NS3, NS4A, NS4B, and NS5) [15]. The E gene, serving as the primary virulence protein of TMUV, occupies a crucial role in virus replication [16]. It contributes to binding with host receptor proteins, facilitating membrane fusion between the virus and the host cells, and determining the virus’s tissue tropism [17]. Moreover, the E protein serves as the primary target for neutralizing antibodies, pivotal for the production of specific antibodies and the elimination of viral infection [17,18,19]. Due to the extensive variability observed, the E gene serves as a frequently utilized locus for genetic variation analyses and molecular epidemiological investigations to ascertain the evolutionary relationships between TMUV strains [20,21,22].

TMUV displays a wide host range, capable of infecting various bird species, including ducks [6,7,8,9,10], geese [23,24], chickens [25,26], sparrows [27], and pigeons [28]. Furthermore, TMUV exhibits sensitivity towards multiple mammalian cell lines, such as BHK-21, Vero, A549, HepG2, HeLa, and SH-SY5Y [29]. Researchers have successfully established a murine model for TMUV infection in the central nervous system (CNS) through intracranial inoculation. This model demonstrates the effective replication of TMUV within the brain, leading to neuronal degeneration, necrosis, and a vigorous inflammatory response, ultimately culminating in mortality [24,30,31]. Additionally, studies have identified the presence of TMUV antibodies and nucleic acids in serum samples and throat swabs obtained from duck farmers [32], indicating the potential for TMUV to emerge as a zoonotic pathogen, capable of inducing diseases in both humans and animals.

TMUV is predominantly found in Asia and exhibits a higher evolutionary rate compared to other members of the Flaviviridae family. Classification based on the E gene reveals the following three distinct clusters: cluster 1, cluster 2, and cluster 3. Research suggests that TMUV initially emerges from cluster 3, with subsequent mutations giving rise to cluster 1 and cluster 2 [20]. Cluster 1 has been primarily identified in Thailand and Malaysia, while cluster 2 predominates in China. Within cluster 2, subcluster 2.2 represents a prominent subgroup [24,25]. Limited information is available regarding cluster 3, as its recognition was first proposed in 2019 [13]. Cluster 3 displays a diverse host range, encompassing ducks, mosquitoes [33,34], chickens [25,26] and geese [35].

There has been a notable increase in the frequency of reports regarding cluster 3 TMUV in recent times [25,26,33,34,35,36,37]. In 2022, a suspected outbreak of TMUV infection emerged at a goose farm in Anhui Province, China. From deceased adult geese, TMUV HQ-22, classified under cluster 3, was successfully isolated. The complete genome of HQ-22 was determined and analyzed to study the virus’ genetic variation. A further pilot study indicated that HQ-22 showcases significant pathogenicity in both goslings and ICR mice. Consequently, it reinforces the notable threat imposed by TMUV on the poultry industry in China, alongside the potential risk of infection and ensuing pathogenicity in mammals.

## 2. Materials and Methods

### 2.1. Clinical Samples and Pathogeny Identification

In September 2022, an outbreak of an infectious disease displaying symptoms such as depression and peritonitis occurred at a goose farm located in Huoqiu County, Lu’an City, Anhui Province, China. Post-mortem examinations conducted on the deceased geese revealed significant pathological alterations, including myocardial hemorrhage, hepatomegaly accompanied by jaundice, intestinal surface hemorrhage, and severe splenic hemorrhage. These findings suggested potential infection by TMUV. To investigate further, brain tissue, spleen tissue, and liver tissue obtained from the affected geese were promptly collected and stored at −80 °C. Subsequently, the collected pathological tissues underwent homogenization in Dulbecco’s modified Eagle’s medium (DMEM, Gibco, Waltham, MA, USA), and the resulting supernatant was filtered through a 0.22-μm filter. This filtered supernatant was then employed for inoculation into 9-day-old goose embryos. Within 3–4 days post-inoculation, the embryos succumbed to the infection, thereby suggesting the presence of the pathogen responsible for the disease. Additionally, urine samples were collected for analysis, and total RNA extraction was performed. RT-PCR was employed to detect the presence of various pathogens, including TMUV, Goose astrivirus (GAstV), Goose parvovirus (GPV), Avian influenza virus (AIV), and Avian reovirus (ARV).

### 2.2. Virus Isolation

Upon reaching 80% confluence, the BHK-21 cells were added to 2 mL of the urine sample mentioned earlier (diluted 1:10 with DMEM medium). The cells were incubated at 37 °C for 2 h in a CO_2_ incubator. Once the incubation was complete, the original medium was removed, and the cells were washed three times with phosphate-buffered saline (PBS). Then, the medium was replaced with fresh DMEM containing 2% fetal bovine serum (FBS, Gibco, Waltham, MA, USA). After 72 h of infection, the supernatant from the culture was collected. The obtained virus was stored in a freezer at −80 °C. To acquire passage viruses, these steps were repeated at least three times.

### 2.3. RNA Extraction and Viruses Detection

According to the FastPure Cell/Tissue Total RNA Isolation Kit (Vazyme, Nanjing, China) instructions, the total RNA was extracted. Then, first-strand cDNA was synthesized by using All-in-one RT EasyMix for qPCR (TOLOBIO, Shanghai, China). The 2 × Taq Master Mix (Dye Plus) was used for RT-PCR of virus detection, as follows: 1 μL cDNA, 0.5 μL of each 10-μM primer, 10 μL mix, and 8 μL ddH_2_O. The absolute quantitative RT-PCR was used to measure the RNA load, as follows: 2 μL cDNA, 0.4 μL of each 10-μM primer, 7.2 μL ddH_2_O, and 10 μL ChamQ Universal SYBR qPCR Master Mix (Vazyme, Nanjing, China). All used primers are shown in Table 1.

### 2.4. Indirect Fluorescent Antibody Assay

To detect the E protein of TMUV, indirect fluorescence assay was used. At 2 days post-infection, cold methanol was used to fix the BHK-21 cells at 4 °C for 20 min. Then, the cells were washed three times with PBS and inoculated with monoclonal antibody (1:1000) against TMUV E protein, which was prepared in our lab [38]. After incubating at 37 °C for 1 h, the BHK-21 cells were washed three time with PBS and then incubated with FITC-labeled Goat Anti-Mouse IgG (1:1000, Solarbio, Beijing, China) at 37 °C for 1h. The BHK-21 cells were washed three times with sterile PBS and then incubated with diamidine phenylindole (DAPI, 1:1000, Solarbio, Beijing, China). Immunofluorescence was observed using a fluorescence microscope (Olympus Inc., Tokyo, Japan).

### 2.5. Whole Genome Sequencing and Genetic Variation Analysis of HQ-22

The full-length genome of HQ-22 was amplified using an overlapping RT-PCR method (primers are shown in Table 2). The 50 μL reaction mixtures were set up with 25 μL 2 × Phanta Max Master Mix (Vazyme, Nanjing, China), 1 μL of each 10-μM primer, 2 μL cDNA, and 21 μL ddH_2_O. The PCR products were purified and cloned into pCE2 TA/Blunt-Zero vector (Vazyme, Nanjing, China), and clones were obtained for sequencing (Tsingke, Nanjing, China) to determine the correct genome information. We used DNAMAN v.6.0 software (Lynnon Biosoft, San Ramon, CA, USA) and Lasergene.v7.1 software (DNAStar, Madison, WI, USA) to splice different fragments with the Clustal W method to obtain the correct full-length genome of HQ-22. The E gene of HQ-22 was aligned with known TMUV strains and the phylogenetic trees of these TMUV E genes were constructed using the neighbor-joining method using MEGA v.11.0 software with bootstrap values calculated from 1000 replicates. The homology analysis of the nucleotide and amino acid sequences of the coding structural protein genes of HQ-22 and other TMUV strains was conducted using Lasergene.v7.1 software (DNAStar) with the Clustal W method.

### 2.6. The Pathogenicity of HQ-22 in Goslings

To investigate HQ-22’s pathogenicity in goslings, 3-day-old goslings (obtained from Hua Ren Agricultural and Livestock Group Co., Ltd., Hefei, China) were used. Prior to experimentation, RT-PCR testing was conducted, confirming that the goslings were free from infection with common goose-origin viruses (GAstV, AIV, ARV, and GPV).

A total of 80 healthy goslings were randomly allocated into two groups, each containing 40 goslings, as follows: the experimental group received an intramuscular injection of HQ-22 virus suspension in the leg muscles at a dosage of 0.5 mL per gosling (virus titer diluted to 10^6^ TCID_50_/0.1 mL), while the control group received an equivalent volume of sterile PBS using the same injection route. The goslings were individually kept, and their body weights were recorded daily. On days 1, 4, 7, 9, 12, and 14 post-infection, three goslings from each group were euthanized for post-mortem examinations, which included virus detection, histopathology (HE staining), and immunohistochemistry (IHC staining) using collected brain and visceral tissues. Any goslings nearing death on the same day were collected for additional research materials.

### 2.7. The Pathogenicity of HQ-22 in Mice

To assess the pathogenicity of HQ-22 in mice, 3-week-old Institute of Cancer Research (ICR) mice were enlisted. Fifty healthy ICR mice were randomly allocated into five groups, with ten in each group. Four groups were subjected to the following different inoculation methods: intracranial (30 µL), intranasal (100 µL), intraperitoneal (100 µL), and intramuscular (100 µL), with a virus titer of 10^6^ TCID_50_/0.1 mL. The control group received an equivalent volume of sterile PBS following the same administration routes. In cases where the mice were near death simultaneously, they were collected for further research purposes. Brain and visceral tissues were obtained from the mice for virus detection, HE staining, and IHC staining.

To analyze the proliferation pattern of HQ-22 in ICR mice, 40 healthy ICR mice were randomly assigned into two groups, consisting of 20 mice each. In the first group, the mice were subjected to intracranial inoculation with HQ-22 following the previously described method and dosage. Subsequently, at 1, 2, 4, 6, and 8 days post-inoculation (dpi), three mice were euthanized for further analysis. In the second group, the mice were inoculated intranasally with HQ-22 using the previously outlined method and dosage. At 1, 3, 6, 9, 12, and 14 dpi, three mice were euthanized for examination. In cases where the mice were nearing death on the same day, they were collected for subsequent research. Brain and visceral tissues were obtained from these mice for virus detection, HE staining, and IHC staining.

### 2.8. Histopathology and Immunohistochemistry

The brains were fixed in 4% formaldehyde in PBS and subsequently embedded in paraffin. Slicing of the embedded tissues into 5-μm-thick sections was carried out using a Leica SM2010R microtome (Leica, Shanghai, China). The sections were then stained with hematoxylin and eosin (H&E) (Beyotime Biotechnology, Shanghai, China). The stained sections were examined under an Eclipse E100 light microscope at 400× *g* magnification (Nikon, Tokyo, Japan). For immunohistochemical (IHC) staining, the fixed tissue sections were blocked and incubated overnight at 4 °C with monoclonal antibody (1:1000) against TMUV E protein, which was prepared in our lab. The sections were then treated with a secondary antibody conjugated to horseradish peroxidase (1:2000 dilution; ab7090, Abcam). After adding diaminobenzidine as the substrate chromagen and counterstaining with hematoxylin, the sections were observed under an Eclipse E100 light microscope (Nikon, Japan).

### 2.9. Statistical Analysis

The statistical analysis was conducted using GraphPad Prism 8.0 software (La Jolla, CA, USA). The results are summarized as means ± SD. A Student t-test was employed to compare two groups, whereas a one-way analysis of variance followed by Tukey’s post hoc test was used for comparisons involving multiple groups. Significance was determined at *p* < 0.05.

## 3. Results

### 3.1. Virus Detection and Virus Isolation

In 2022, a goose farm located in Anhui Province, China, experienced an outbreak of an infectious disease exhibiting symptoms of a depressed mental state and peritonitis. The affected geese were found to be adult and displayed noticeable pathological alterations, including myocardial hemorrhage, hepatomegaly with jaundice, intestinal surface bleeding, and severe splenic hemorrhage (Figure 1A). These observations suggested a potential infection with TMUV. Liver, spleen, and heart tissues from the affected geese were collected and homogenized with DMEM, followed by filtration through a 0.22-μm filter. The resulting supernatant was then inoculated into 9-day-old goose embryos (*n* = 5), with respective control groups in place. By utilizing RT-PCR, it was confirmed that the goose embryos were negative for common goose-origin viral infections. All of the goose embryos succumbed to the viral fluid at 3 dpi, presenting severe congestion on the surface of the infected embryos (Figure 1B). Total RNA was extracted from collected urine samples and subjected to RT-PCR to detect GAstV, GPV, AIV, TMUV, and ARV. The results revealed that only TMUV was present (Figure 1C). Subsequently, the urine fluid containing the virus was used to infect the BHK-21 cells. Immunofluorescence analysis was performed using a monoclonal antibody specific to the E domain III of TMUV, which was developed in our laboratory. This analysis revealed the presence of abundant green fluorescence signals localized in the cytoplasm of the BHK-21 cells (Figure 1D). Moreover, the infected BHK-21 cells displayed distinct pathological changes, such as wrinkling and rupturing, at 60 h post-infection (Figure 1E). The TMUV strain successfully replicated in the BHK-21 cells, with a viral titer reaching 10^6.0^ TCID_50_/0.1 mL, thereby confirming the successful isolation of the TMUV strain from the goose. This isolated strain was designated as HQ-22.

### 3.2. Phylogenetic and Evolutionary Analyses of TMUV HQ-22

We utilized a laboratory-designed set of 10 sequencing primers, which strategically overlap with each other and ensure the coverage of the entire TMUV genome. Through this approach, we successfully amplified 10 target fragments. These purified target fragments underwent sequencing analysis, and the resulting sequences were aligned and assembled to reconstruct the complete genome of TMUV HQ-22, which spans 10,994 bp (GenBank No. OR909676).

The phylogenetic analysis of the HQ-22 isolate—focusing on the key virulence gene, the E gene of TMUV—was performed using MEGA v.11.0 software. The results have demonstrated that the HQ-22 isolate was classified within cluster 3 and exhibited the closest genetic relationship to the mosquito-derived P73_TH_2019 strain and the chicken-derived CTLN strain (Figure 2). The reference indicates that the P73_TH_2019 strain and three other isolates reported in Thailand in 1992 and 2002 both belong to cluster 3, indicating that cluster 3 isolates have been spreading in Thailand for many years [33].

The MegAlign v.11.0 software was utilized to conduct an analysis evaluating the level of nucleotide and amino acid identity among the ORF and structural protein genes of the HQ-22 isolate and representatives of cluster 3 TMUV strains, as well as representative strains from other clusters. The findings have revealed that the HQ-22 isolate exhibited the highest degree of nucleotide and amino acid identity with cluster 3 strains, reaching an impressive 99.2% and 99.8%, respectively, thus indicating a considerable level of identity. In contrast, when compared to the TMUV strains from the other clusters, the HQ-22 isolate only shared a maximum of 89.8% nucleotide identity and 98.1% amino acid identity (Table 3).

The HQ-22 isolate demonstrated a substantial degree of identity with the TMUV CTLN strain, which originated from chickens and belonged to cluster 3. Specifically, the amino acid identity between the HQ-22 and the CTLN strains was 99.2% for the C gene, 100% for the PrM gene, and 99.8% for the E gene. Upon comparing the amino acid sequences of the E gene encoded by strains from various clusters using MEGA v.11.0 software (Figure 3), we discovered that the 157th amino acid in the E protein, typically encoded by cluster 3 strains, was valine (V). However, the HQ-22 strain displayed alanine (A) at the 157th position, identical to the cluster 2 strains. We observed that HQ-22, P49_TH_2019, P73_TH_2019, CTLN, and GX2021 displayed isoleucine (I) at the 358th position, while the other strains displayed valine (V). Additionally, we observed that cluster 3 shared more identical amino acid sites with both cluster TMUV and cluster 1, indicating that the TMUV strains within cluster TMUV may serve as ancestral forms of all of the TMUVs, gradually diverging into cluster 3, cluster 1, and cluster 2. The mutation in the amino acid encoded by the E gene in HQ-22 may influence its tissue tropism and pathogenicity in animals. This suggests that the ongoing mutation and evolution of TMUV may present new challenges for preventing and controlling TMUV in the coming years.

### 3.3. TMUV HQ-22 Infection in Goslings

A group of goslings was infected via intramuscular injection of 0.5 mL of HQ-22 (10^6.0^ TCID_50_/0.1 mL), while the control group received an equal amount of sterile PBS using the same method. Following the viral infection, the goslings in the infected group displayed evident neurological symptoms, including depression, decreased appetite, and ataxia (Figure 4A). The postmortem examination revealed severe brain tissue bleeding and congestion among the infected goslings (Figure 4B). The overall mortality rate of the goslings reached 60% by 14 dpi (Figure 4C). The brains and internal organs of the deceased goslings in the infected group were subjected to RT-qPCR analysis for TMUV amplification, which demonstrated the presence of the virus in all of the tissues. The highest viral load was observed in the brain tissue, reaching 10^6^ copies/μg RNA (Figure 4D,E). Viral amplification peaked in the brain tissue at 7 dpi, as observed by monitoring the virus levels at different time points (Figure 4D,E). The HE staining of the brain tissues from the infected group of goslings revealed significant vascular cuffing and lymphocyte infiltration (Figure 4F). The IHC staining, employing a TMUV-E-specific monoclonal antibody developed in our laboratory as the primary antibody, exhibited prominent positive antigen signals of the virus in the brain tissues of the infected group of goslings (Figure 4G). This indicated that viral replication extensively occurred in the brain tissues of the infected goslings, leading to CNS damage and eventual mortality.

### 3.4. TMUV HQ-22 Infection in Mice

To investigate the pathogenicity of TMUV in mammals, mice have been commonly employed as reliable pathological models. For the current study, three-week-old ICR mice were chosen as the animal model for infection, aiming to explore the pathogenicity of TMUV in mammals. The mice were infected using various routes, namely intracranial inoculation, intranasal inoculation, intraperitoneal injection, and intramuscular injection. The infected mice were closely observed for 14 days, and the results showed that the mice infected through intracranial inoculation exhibited the highest sensitivity. In the later stages of infection, all of the mice in this group displayed evident neurological symptoms like decreased appetite, whole-body tremors, and hind-limb paralysis. The postmortem examination revealed notable brain hemorrhage in the mice (Figure 5A), and all of the mice succumbed to the infection by 9 dpi (Figure 5C). Additionally, the mice inoculated with HQ-22 via the intranasal route did not present significant neurological symptoms in the majority of cases, except for slight weight loss. However, it is worth mentioning that 10% of the individuals also displayed severe neurological symptoms and died, with the postmortem examination revealing significant brain hemorrhage in these mice as well (Figure 5B). However, it was found that HQ-22 demonstrated an incapability of infecting mice via the intraperitoneal or intramuscular injection routes, with no significant differences observed compared to the control group in terms of mice infected through these routes. Notably, the HE staining revealed characteristic symptoms of viral encephalitis in the mice from the intracranial and intranasal injection groups, including tissue congestion, vascular cuffing, and lymphocyte infiltration (Figure 5D). Additionally, the IHC staining demonstrated robust viral antigen signals in the brain tissues of the mice from both of the injection routes (Figure 5E). The RT-qPCR results indicated that virus replication was only prominent in the brains of the mice, and viral nucleic acid was barely detectable in the internal organs and blood. Throughout the infection period, the viral presence was consistently detected in the mouse brain tissues in the intracranial injection group, peaking at 10^6^ copies/µg RNA. However, in the intranasal infection group, the virus was only detectable in the brain at 6 dpi and subsequently declined to low levels by 14 dpi (Figure 5F).

## 4. Discussion

TMUV is a novel mosquito-borne flavivirus that first surfaced in Malaysia in 1955, with sporadic reports involving mosquitoes in subsequent years [4,22]. However, there were no indications of extensive transmission or animal infection by TMUV. Notably, in 2010, TMUV outbreaks emerged in various regions of China, ultimately escalating nationwide [22,33,39,40]. The virus continues to undergo rapid evolution and mutation [20]. TMUV predominantly impacts the ovaries, spleen, and brain tissues of ducks, resulting in a significant reduction in egg production, a compromised immune system, and mortality, necessitating the culling of meat ducks [16,39,41,42]. Consequently, the duck farming industry in China has suffered substantial economic losses due to this virus.

TMUV can be categorized into three distinct clusters based on the E gene, namely cluster 1, cluster 2, and cluster 3. The majority of TMUV strains identified in China and Southeast Asian countries belong to cluster 2. Cluster 3, a relatively recent phenomenon, has emerged in recent years. Strains such as DK/TH/CU-56, isolated from Thailand in 2016, and SD14, isolated from China in 2014, are representative examples of this cluster [13]. P49_TH_2019, P73_TH_2019 and three other TMUV strains isolated in Thailand in 1992 and 2002 all belong to cluster 3 [34]. The available evidence supports the notion that cluster 3 is the most primitive, indicating that cluster 1 and cluster 2 evolved from cluster 3 through mutation [20]. The TMUV strains belonging to cluster 3 have been detected in a diverse range of hosts, including mosquitoes [34,40], ducks [13], chickens [25,26], sparrows [27], and geese [35]. Notably, our research group has successfully isolated cluster 3 TMUV from geese as well. Recently, an increasing number of cluster 3 TMUV strains have been identified in various hosts, indicating a growing prevalence of cluster 3 TMUV in China. This underscores the importance of recognizing the potential economic losses and biosafety concerns associated with cluster 3 TMUV.

TMUV, a mosquito-borne virus, is primarily transmitted through mosquito bites [22,38]. However, intriguingly, studies have revealed that TMUV can persist even during the winter, when mosquito activity is minimal [43]. Li conducted research indicating the presence of TMUV in aerosols, which can be transmitted to birds through respiratory routes. Yan’s study demonstrated high viral levels detected in the trachea of infected ducks, enabling direct transmission to cohabiting birds [44]. Furthermore, Yan identified a mutation (E_S156P_) in the E protein, resulting in the loss of N154 glycosylation, reduced tissue tropism in ducks, and decreased inter-avian transmission ability [45]. Notably, the HQ-22 strain isolated in our study carries a unique amino acid mutation (E_A157V_) at position 157 of the E protein, specific to cluster 2. Cluster 2 currently represents the most prevalent TMUV strains in China, primarily affecting waterfowl, such as ducks. Conversely, cluster 3, an ancestral cluster, has been predominantly spreading through mosquitoes. We observed that HQ-22, P49_TH_2019, P73_TH_2019, CTLN (isolated in 2020), and GX2021 displayed isoleucine (I) at the 358th position, while other strains displayed valine (V). These strains are clustered into the same branch in the phylogenetic analysis (Figure 2), suggesting that the cluster 3 strains isolated in China in recent years may be from Thailand. These findings underscore the ongoing evolution of TMUV in different hosts, expanding its range of host adaptation from mosquitoes to various avian species, including ducks, chickens, and geese, with a potential risk of mammalian infection.

Yu’s research has revealed that TMUV’s CTLN strain, belonging to cluster 3 and originating from chickens, can proliferate effectively in BHK-21, DEF, and CEF cells. In particular, it demonstrates higher replication levels in the C6/36 cells derived from mosquitoes compared to the TMUV strains from cluster 2 [25]. Moreover, Yan’s investigation has shown that chickens can be infected with cluster 3 TMUV and exhibit symptomatic disease when intranasally or intramuscularly inoculated; however, in groups of chickens in contact with these, no disease transmission or symptoms were observed [26], suggesting that TMUV infection in chickens is primarily transmitted via mosquito bites. It is noteworthy that many members of the Flavivirus genus, including JEV, WNV, DENV, ZIKV, and TBEV, have been recognized to induce viral encephalitis in the central nervous system of mammals, including humans [46,47,48,49]. Prior studies have demonstrated that mice can develop symptoms like wasting, circling behavior, and hindlimb paralysis following intracranial inoculation with TMUV. In severe cases, this can lead to mouse mortality, with a notably higher viral load observed in the brain tissue compared to other visceral tissues [24]. However, when a different inoculation method, such as intramuscular injection or intraperitoneal injection, is employed, no discernible disease or changes are observed in mice [30]. This suggests that TMUV is unlikely to effectively infect mammals through mosquito bites in a natural environment. The experiments conducted by our research group have demonstrated that the intracranial inoculation of TMUV consistently triggers neurological symptoms and mortality in mice. Similarly, the intranasal inoculation of mice leads to weight loss, substantial brain tissue hemorrhage upon necropsy, and the presence of the virus in the brain tissue. Severe cases can also result in disease and death. Conversely, mice in the peripheral infection groups, including the intraperitoneal and intramuscular injection groups, exhibit no discernible clinical changes or detectable viral presence. These findings suggest that TMUV, in its natural environment, has evolved mechanisms to infect the CNS in mice via routes other than intracranial inoculation. This may lead to the development of severe viral encephalitis in the mammalian CNS, similar to other members of the Flavivirus genus.

## 5. Conclusions

In summary, we isolated a TMUV strain, TMUV HQ-22, from geese that belongs to cluster 3 and exhibits significant pathogenicity in both goslings and ICR mice. In recent years, there has been a notable surge in reports regarding TMUV from cluster 3, surpassing its previous restriction to waterfowl hosts, such as ducks. This serves as a reminder that a less-reported cluster of TMUVs is emerging, presenting novel challenges in prevention and control. Furthermore, it is imperative for the public to acknowledge and address the potential threat of TMUV infecting mammals with utmost seriousness.

## Figures and Tables

**Figure 1 viruses-15-02449-f001:**
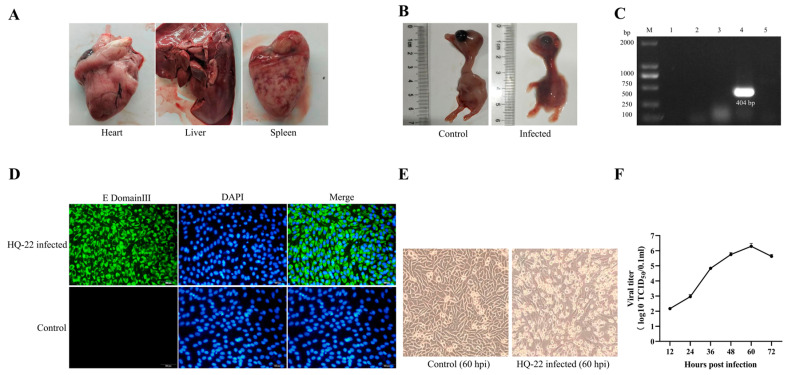
Isolation and identification of the virus. (**A**) Pathological changes observed during necropsy of diseased geese. (**B**) Death of goose embryos with hemorrhage on the embryo surface after inoculation with HQ-22. (**C**) RT-PCR results of urine samples: (1–5) represent samples tested for GAstV, GPV, AIV, TMUV, and ARV. (**D**) Immunofluorescence experiment showing the distribution of HQ-22’s E protein in BHK-21 cells, 400× *g* magnification. (**E**) Infected BHK-21 cells exhibiting significant cellular lesions at 60 h post-infection, 400× *g* magnification. (**F**) Growth curve of HQ-22 on BHK-21 cells.

**Figure 2 viruses-15-02449-f002:**
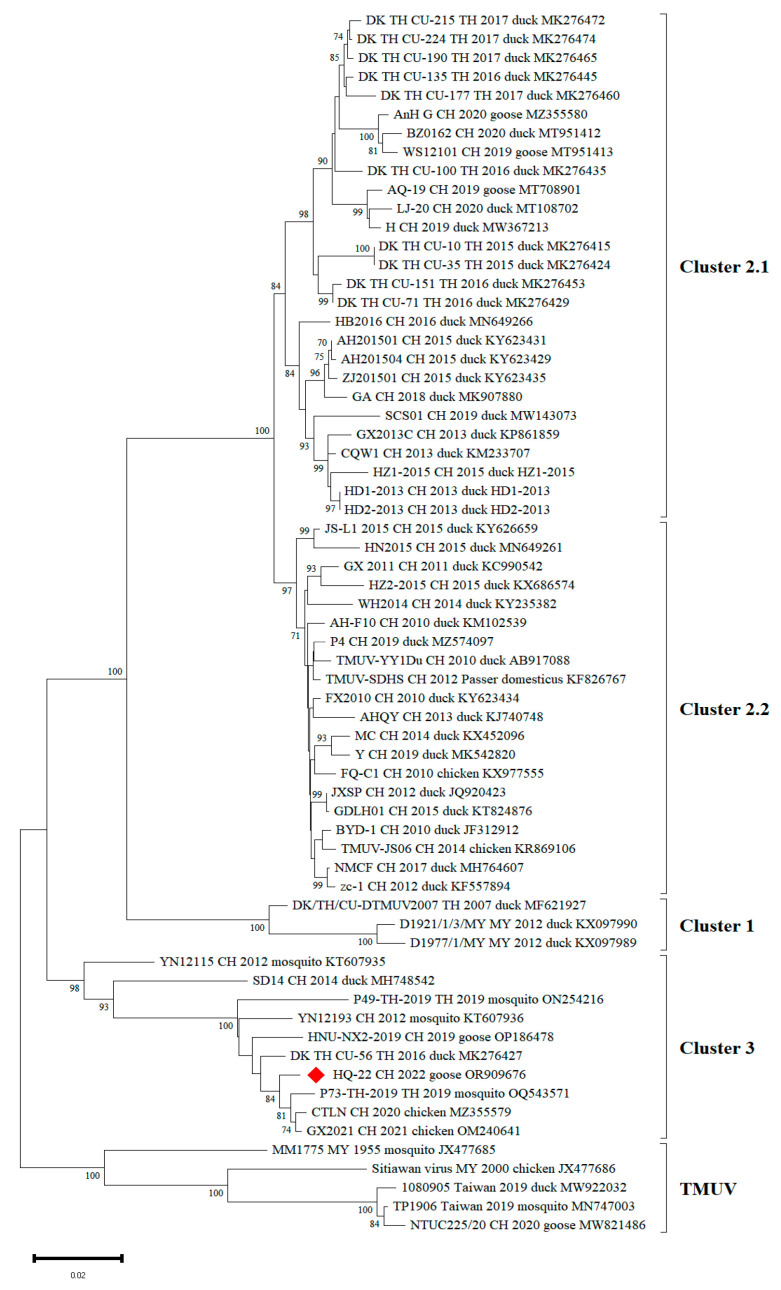
Phylogenetic analysis based on the Tembusu virus E gene. The phylogenetic tree was constructed based on the TMUV E gene (1503 bp) using the NJ method implemented on MEGA v.11.0. Bootstrap values are shown on the nodes. The phylogenetic analysis displays the strain name, origin, year of isolation, host, and GenBank accession numbers for each virus strain.

**Figure 3 viruses-15-02449-f003:**
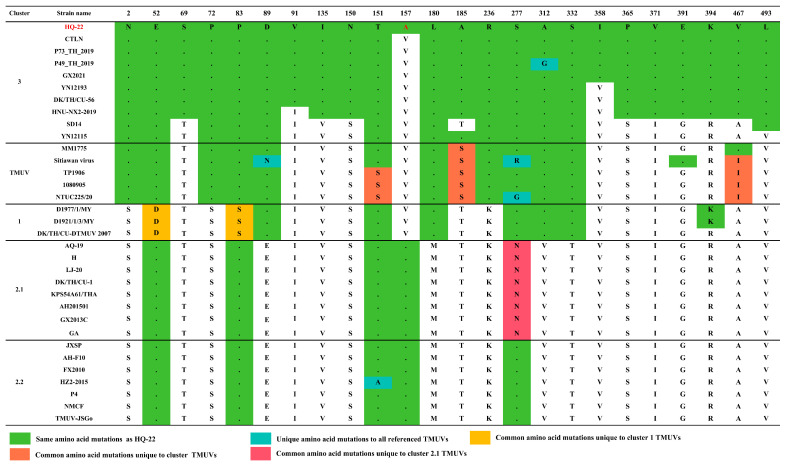
Summary of amino acid differences in the E protein between HQ-22 and selected TMUV isolates from other clusters.

**Figure 4 viruses-15-02449-f004:**
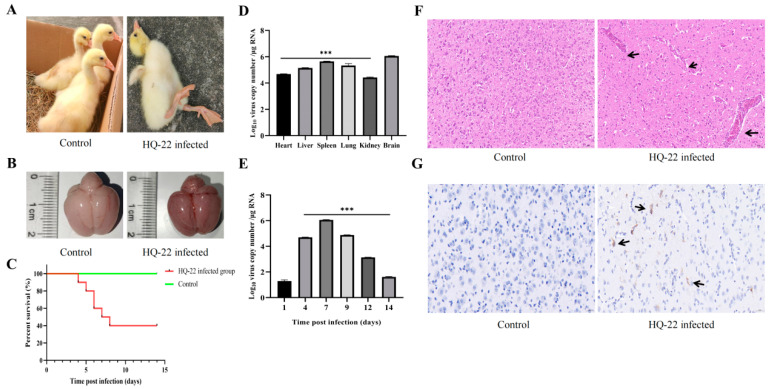
Clinical characteristics, pathological changes, and virus detection in goslings infected with HQ-22. (**A**) Goslings in the infected group showed stunted growth, ataxia, and an inability to stand. (**B**) Severe brain hemorrhage was observed in goslings in the infected group. (**C**) Record of goslings’ survival status. (**D**) Distribution of the virus within deceased goslings, compared with viral load in the brain. (**E**) Viral amplification trend in gosling brains, compared to viral load in the brain at 1 dpi. Data are shown as means ± SD of three separate experiments. * *p* < 0.05, ** *p* < 0.01, *** *p* < 0.001. (**F**) HE staining was used to observe the gosling brain tissue (100× *g* magnification), cerebral hemorrhage, and perivascular cuffing (arrows). (**G**) Immunohistochemistry was applied to observe the distribution of viral antigens in the brain tissue. Positive signals appear as brownish-yellow areas (arrows), and the primary antibody used was an anti-TMUV-E monoclonal antibody (200× *g* magnification).

**Figure 5 viruses-15-02449-f005:**
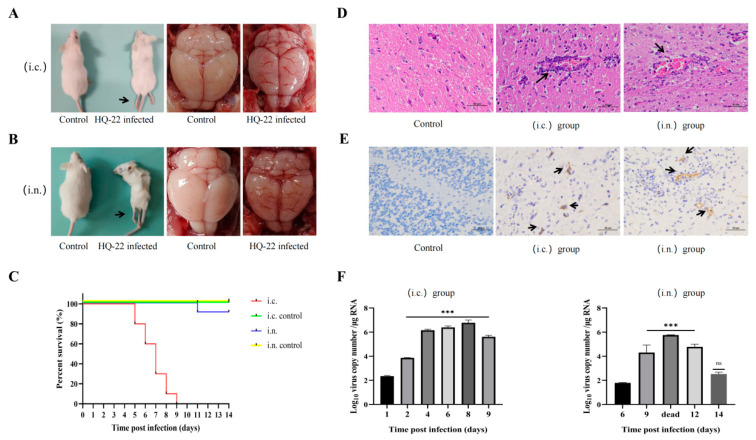
Clinical features, pathological changes, and virus detection in ICR mice infected with HQ-22. (**A**) Representative image of mice from the intracranial (i.c.) injection group showing reduced body size, hindlimb paralysis, and brain hemorrhage. (**B**) Representative image of mice from the intranasal (i.n.) injection group showing reduced body size, hindlimb paralysis, and brain hemorrhage. (**C**) Survival curve of infected mice. (**D**) HE staining of mouse brain tissues showing pathological changes (400× *g* magnification), cerebral hemorrhage, and perivascular cuffing (arrows). (**E**) IHC staining showing the presence of viral antigens in the brains of infected mice (400× *g* magnification). Positive signals appear as brownish-yellow areas (arrows). (**F**) Viral amplification trend in mouse brains infected with HQ-22, compared to viral content in the brains at 1 dpi or 6 dpi. Data are presented as mean ± SD of three separate experiments. * *p* < 0.05, ** *p* < 0.01, *** *p* < 0.001, ns *p* ≥ 0.05.

**Table 1 viruses-15-02449-t001:** Primers used for virus detection.

Primer	Sequence (5′-3′)	Annealing Temperature	Size
TMUV-F	CTGATGGCTTTGGTCCTGTTC	52 °C	404 bp
TMUV-R	ACACCTATTCTCACCACATCTAAC		
TMUV-RT-qPCR-F	ACACCTATTCTCACCACATCTAAC	60 °C	180 bp
TMUV-RT-qPCR-R	TAACAAGTGGCAGAGCAAGGG		
GAstV-F	AGAAGGTGCGGAAGAGTGGTATGA	55 °C	300 bp
GAstV-R	GCGAAGAGTGCGTAAGAGGTTGT		
GPV-F	CCAAGCTACAACAACCACATCTAC	54 °C	375 bp
GPV-R	CTGCGGCAGGGCATAGACATCCGAC		
AIV-F	TTCTAACCCAGGTCGAAAC	51 °C	229 bp
AIV-R	AAGCCTCTACGCTGCACTCC		
ARV-F	TCTCGAGATCTAACTAGATCTGA	55 °C	519 bp
ARV-R	CGTGTCCAACACCAAGTAAACAC		

**Table 2 viruses-15-02449-t002:** Primers used for amplification of the genome of TMUV HQ-22.

Primer	Sequence (5′-3′)	Annealing Temperature	Size
T1-F	AGAAGTCCATCTGTGTGAAC	51 °C	1426 bp
T1-R	GGCTGAATAATTATGGTAG		
T2-F	TTGTTTGGAAAGGGGAGC	55 °C	984 bp
T2-R	TACACCCCCGACTGAGCCAA		
T3-F	TGGTTGCTTTGGGTGAC	51 °C	1165 bp
T3-R	CCACTCGCTGTTGTTGTC		
T4-F	AATAGACTTCGACTACTGCC	51 °C	939 bp
T4-R	AAAGCCTCACTGACTGG		
T5-F	GTCCTTTGGTGTTTGCGGGTTTGC	58 °C	1432 bp
T5-R	GAGTCCGGAAAGGCGTCAGTTGTG		
T6-F	CAAAGGTGGAACTGGGAGA	57 °C	1002 bp
T6-R	GAGCGAAGTGGTCAGGAAG		
T7-F	AGGATTTTGCGAGTGG	50 °C	1213 bp
T7-R	TGGAGGTTCCGAGATAT		
T8-F	GCCGTATCTGGAATGCAACTACGGC	60 °C	1434 bp
T8-R	CGACAAGACTCCAGAATTCTGGGTC		
T9-F	GCCATGTTTGAGGAGC	50 °C	1462 bp
T9-R	AGCTTTCAATGGGTTTG		
T10-F	CCCAATTATGCAGATCA	59 °C	1142 bp
T10-R	AGACTCTGTGTTCTACCAC		

**Table 3 viruses-15-02449-t003:** Identity analysis of nucleotide and amino acid sequences encoded by ORF and structural protein genes (C, PrM, and E) between the isolate and strains from different clusters.

Reference TMUVs	Accession No.	Year of Collection	Location	Host	(ORF) Nucleotide Identity	(ORF) Amino Acid Identity	(C) Nucleotide Identity	(C) Amino Acid Identity	(PrM) Nucleotide Identity	(PrM) Amino Acid Identity	(E) Nucleotide Identity	(E) Amino Acid Identity
**Cluster 3**												
CTLN	MZ355579	2020	China	chicken	99.2%	99.8%	98.6%	99.2%	99.4%	100%	99.1%	99.8%
GX 2021	OM240641	2021	China	chicken	99.1%	99.6%	98.1%	99.2%	99.2%	100%	99.1%	99.4%
HNU-NX2-2019	OP186478	2019	China	goose	97.7%	99.3%	98.1%	99.2%	97.2%	100%	97.7%	99.0%
YN12193	KT607936	2012	China	mosquito	97.3%	99.3%	97.8%	98.3%	97.4%	100%	97.5%	99.4%
YN12115	KT607935	2012	China	mosquito	93.1%	98.2%	95.3%	96.7%	93.8%	98.8%	93.8%	97.2%
SD14	MH748542	2014	China	duck	93.7%	97.6%	93.6%	95.0%	94.2%	96.4%	93.3%	95.0%
DK/TH/CU-56	MK276427	2016	Thailand	duck	—	—	—	—	—	—	98.3%	99.4%
P49_TH_2019	ON254216	2019	Thailand	mosquito	—	—	—	—	—	—	96.2%	97.4%
P73_TH_2019	OQ543571	2019	Thailand	mosquito	—	—	—	—	—	—	98.6%	99.0%
**TMUV**												
MM1775	JX477685	1955	Malasia	mosquito	89.6%	97.5%	93.9%	97.5%	91.6%	100%	88.8%	96.8%
Sitiawan virus	JX477686	2000	Malasia	chicken	88.1%	97.2%	91.9%	96.7%	86.2%	97.0%	87.2%	96.8%
TP1906	MN747003	2019	China	mosquito	87.4%	98.1%	91.4%	97.5%	87.0%	98.8%	86.5%	96.8%
1080905	MW922032	2019	China	duck	87.3%	97.1%	91.1%	96.7%	86.6%	98.2%	86.2%	96.2%
NTUC225/20	MW821486	2020	China	goose	87.3%	97.0%	91.1%	96.7%	86.8%	98.2%	86.3%	96.6%
**Cluster 1**												
D1977/1/MY	KX097989	2012	China	duck	89.6%	96.4%	91.9%	95.0%	90.0%	97.5%	89.5%	95.8%
D1921/1/3/MY	KX097990	2012	Malasia	duck	89.6%	96.4%	91.7%	95.0%	89.8%	97.6%	89.3%	95.8%
DK/TH/CU-DTMUV2007	MF621927	2007	Thailand	duck	89.8%	96.6%	91.9%	94.2%	89.0%	97.6%	89.9%	96.4%
**Cluster 2.1**												
AQ-19	MT708901	2019	China	goose	88.9%	96.4%	90.3%	94.2%	88.0%	97.6%	88.1%	95.6%
H	MT108702	2019	China	duck	88.9%	96.4%	90.6%	94.2%	88.0%	97.6%	88.1%	95.8%
LJ-20	MW367213	2020	China	duck	88.8%	96.3%	90.0%	93.3%	87.6%	97.6%	88.2%	95.8%
DK/TH/CU-1	KR061333	2013	Thailand	mosquito	89.4%	96.7%	90.6%	94.2%	88.6%	97.6%	89.0%	95.8%
KPS54A61/THA	KF573582	2013	Thailand	duck	89.4%	96.6%	91.1%	94.2%	89.6%	97.6%	88.9%	95.8%
AH201501	KY623431	2015	China	duck	89.1%	96.6%	91.4%	94.2%	88.2%	97.6%	88.9%	95.8%
GX2013C	KP861859	2013	China	duck	89.3%	96.5%	91.1%	93.3%	89.0%	97.6%	88.6%	95.0%
GA	MK907880	2018	China	duck	88.9%	96.3%	91.1%	94.2%	87.8%	96.4%	88.5%	95.0%
**Cluster 2.2**												
JXSP	JQ920423	2012	China	duck	89.5%	96.8%	90.8%	94.2%	89.0%	97.6%	89.5%	96.0%
AH-F10	KM102539	2010	China	duck	89.5%	96.7%	90.6%	93.3%	89.2%	97.6%	89.2%	96.2%
FX2010	KY623434	2010	China	duck	89.6%	96.8%	91.4%	94.2%	89.6%	97.6%	89.5%	96.0%
HZ2-2015	KX686574	2015	China	duck	89.0%	96.5%	91.7%	93.3%	88.0%	97.0%	88.8%	95.6%
P4	MZ574097	2019	China	duck	89.5%	96.7%	91.1%	94.2%	89.2%	97.6%	89.4%	96.0%
NMCF	MH764607	2017	China	duck	89.6%	96.8%	90.6%	94.2%	88.8%	97.6%	89.1%	95.8%
TMUV-JSGo	AB917090	2012	China	goose	89.4%	96.8%	90.8%	94.2%	88.8%	97.6%	89.5%	96.2%

## Data Availability

The raw data used and analyzed during the current study are available from the corresponding author upon reasonable request.

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
