# Peer review of "Pathogenicity and Interspecies Transmission of Cluster 3 Tembusu Virus Strain TMUV HQ-22 Isolated from Geese"

_viruses, 2023, doi:10.3390/v15122449_

Round 1
Reviewer 1 Report
Comments and Suggestions for Authors
This manuscript by Yang et al, describes the isolation of a new Tembusu virus in geese in China. The isolate was causing disease in the poultry industry. They isolated the virus and sequenced, confirming it is a TMUV virus that falls within the Cluster 3 lineage. It has a unique change in the envelope protein that may suggest it represent a virus in Cluster 3 that gave rise to Cluster 2. Animal studies in both geese and mice demonstrate the virus causes severe pathology in geese, but only causes disease in mice if directly infecting the brain. The study is well written and highlights an emerging pathogen important to the poultry industry.
Suggestions to improve the manuscript:
Figure 1 D and E: Zoom in on the center of the pictures so one can see the cells better.
Minor:
Line 111: I do not think embryo death after injection “confirming the presence of the pathogen. . .” when no purification or isolation was first performed. Should say “suggesting the presence of the pathogen”
Line 119: methods should be in past tense, replace “should be” with “were”
Line 129: Please include information about the antibodies used.
Line 231-260: replace “homology” with “identity” or “similarity”
Author Response
Point 1: Figure 1 D and E: Zoom in on the center of the pictures so one can see the cells better.
Response 1: Thank you very much for your comment. We have enlarged figures 1D and E appropriately to better see the cells. Please check it in our revised manuscript.(Line 241, 246-247)
Point 2: Line 111: I do not think embryo death after injection “confirming the presence of the pathogen. . .” when no purification or isolation was first performed. Should say “suggesting the presence of the pathogen”.
Response 2: We gratefully appreciate for your valuable comments. According to your valuable suggestions, we have revised in the manuscript. (Line 111)
Point 3: Line 119: methods should be in past tense, replace “should be” with “were”.
Response 3: Thank you for your careful advice. Considering your valuable suggestion, we have revised in the manuscript. (Line 118)
Point 4: Line 129: Please include information about the antibodies used.
Response 4: We gratefully appreciate for your valuable comments. We have added the information about the antibodies and dye used. (Line 136-143)
Point 5: Line 231-260: replace “homology” with “identity” or “similarity”.
Response 5: Thanks for your suggestion. According to your valuable suggestions, we have revised in the manuscript. (Line 264-273)
Reviewer 2 Report
Comments and Suggestions for Authors
Summary
The authors of the paper focus their study on the emergence of a new cluster (cluster 3) of the Tembusu virus in China with (i) a genome and proteins residues phylogenetic comparison (ii) pathogenesis report in geese and mice.
Overall impression/ Broad comments
The research treated in this article revealed the recent emergence of a new cluster of TMUV, the cluster 3. This cluster present specificity compares with other former clusters. The importance of a specific amino-acid residue substitution in the Envelope protein of TMUV are reported. The specify in pathogenesis is also reported.
Authors used only one strain of TMUV cluster-3 for pathogenesis investigation in this study and develop a phylogenetic approach to compare with other described clusters.
Despite strength points, there are lack of information to consider this manuscript for publication in its current state. I’ve some suggestion to improve the manuscript for a publication of a revised version.
Major comments
Line 56-57: please add reference for this statement.
Line 58: please add reference for this statement.
Line 83: a very recent publication, Hamel 2023, “Identification of the Tembusu Virus in Mosquitoes inNorthern Thailand ”, described the presence of the TMUV cluster 3 in mosquitoes in North Thailand. This important paper is lacking in this manuscript. Please add this reference.
Line 87 to line 97. Please rephrase this part. Here it’s the “introduction part” not the result or discussion part. Introduce the purpose and the question of the study and what you did to answer the question, but without presenting results of the study itself yet. Keep results for results part.
Materials and Methods
Line 115-116. This sentence can be deleted as it’s a result and author already addressed this point in the dedicated section.
From Line 117 to Line 192: authors used in all this section the abbreviation “DTMUV” to describe their virus. In all the document, author used “TMUV” to describe their virus. Please, homogenate and used only “TMUV”.
Line 136; Please short description of the technic to fully sequence the genome.
Line 137. Describe the alignment: software, method etc..
Line 138: “… fragments and colonies”. Please specify what the term "colonies" refers to?
Line 140-144: please add information about what is the model used for phylogenetic analysis. How the author determines this model?
Line 144: according to the phylogenetic tree, authors used 64 sequences to build it. Please add a table with all sequence used including : Name/accession number/year of isolation/Country
Line 148-149. Authors said they performed a RT-PCR to screen the presence a set of viruses. Please add a description of the PCR for these viruses including primers sequences used, length of amplicon for each virus, PCR protocol etc… This description can be added to the section 2.3 of the manuscript.
Line 149. Remove “such as” as only these 4 viruses with the TMUV were tested (according to the figure 1C. Otherwise please, specify all the tested viruses.
Results
Line 204. Remove “unfortunately”
Line 210-211. Add reference for the antibody if any.
Figure 1-A. There are only pictures of infected geese organs. Please add pictures of control animal for each organ to compare the physio pathological differences. Like in Fig1-B.
Line 226: add a table of primers sequences.
Line 229/230. Which programs were used to assemble fragment and for the alignment (add in section M&M).
Line 230: Add the access number in the text.
Line 234/236: “Notably … it’s not a result. Move this sentence in discussion section,
Section 3.2 and Table 1. Author didn’t add in their analyzes the 2 sequences isolated in 2019 and recently described by Hamel et 2023 (Viruses 2023) in North Thailand.
This point is an important point and an important lack in the manuscript as the TMUV cluster 3 isolated in North Thailand represent the latest isolate in this country. Authors also forget to specify that 3 other isolates were reported in Thailand in 1992 and 2002.
Please add isolates from Hamel et al in the manuscript and include the sequences to the phylogenetic analysis (add the reference too).
Figure 2. Add the latest strain of TMUV cluster 3 described by Hamel et all 2023 in the phylogenetic tree.
Line 252-261. Please add in this part the latest strain of TMUV cluster 3 described by Hamel et all (Viruses 2023). According to this publication, the amino acid in position 358 of the two sequences is a isoleucine (I) residue. This is the same residue than in strain HQ-22, CTLN and GX2021, it’s so very important to discussion this homology compared to the other Cluster 3 strains presenting a valine (V) on position 358.
Figure 3 Add in the amino acid differences the latest strain of TMUV cluster 3 described by Hamel et all 2023
Line 296. Please add information about the black arrows in the figure legend.
Line 300. Why authors selected ICR mice for this challenge?
Figure 5 legend. Add the meaning of ic and in in the figure legend.
Discussion part
Line 345 “…evolution and mutation.” Add reference (I think ref 20)
Line 355 add reference of strain isolated in Thailand in 2019 (ref Hamel et al 2023)
Line 358 add Hamel et al 2023 for mosquito isolations.
Line 376 to 380. Authors suggest that the substitution A157V have an impact for tissue tropism. But there isn’t any result in their study that support this suggestion. Author use only the strain HQ-22 to infect geese and mice without any comparison with infection with other Cluster 2 and cluster 3 strain. So, it’s not possible for the author to suggest any impact of this amino-acid in the tropism and pathogenesis. Please remove this or rephrase with more with more caution.
Line 381 or in other place in discussion: I invite the author to discuss about the amino-acid substitution in position 358 (I358V) with comparison with strain from North Thailand isolated in 2019.
Line 404. Remove “subcutaneous” as this route of injection was not used described in the experiments reported in this manuscript.
Line 405-407. It’s not clear. Why authors said the virus evolved and develop mechanism tho breach the blood-brain barrier? Which results support this evolution as former paper describe the existence of airborne route? Here authors did intranasal injection, but there is a possibility to enter in contact and penetrate via olfactive nerves directly. More investigation is needed to support the evolution of infectious mechanism to pass through the BBB. I invite the author to be more cautious in this part of their discussion and to rephrase.
Line 407-409. Rephrase because TMUV already leads to severe viral encephalitis in bird.
Conclusion
Line 415: “new cluster.. is emerging”. In the manuscript, authors wrote that this cluster 3 is an ancestor of other clusters 1 and 2, so may I know if they consider Cluster 3 as a “new cluster” or if they think this cluster wasn’t reported previously?
Minor issues and comments
Please, remove the reference number of every product in Materials and Methods part and homogenate the presentation of reference of product used in this study.
Line 53. “.. a solitary open…” not a really good English to describe “ a single open reading frame”
Line 63: you can also add the reference (37) here.
Line 128 and 129: please describe shortly the RNA extraction RT-PCR. Add in a supplementary table set of primers used in this study for the TMUV detection. Describe shortly the fluorescent observation assay.
Line 147: I think it’s better to write “ were used” instead of “were utilized”
Line 375 “…cluster3, an ancestral strain, …” used the conditional tense not directly “an ancestral strain”.
Conclusion
Line 411. “our study isolated a…”, I think it’s not correct as “the study” didn’t do the isolation itself. Authors did. Please rephrase.
Comments on the Quality of English LanguageThe manuscript is well written, with a few typos. The English is average with writing that is sometimes more literary than scientific.
Author Response
Point 1: Line 56-57: please add reference for this statement.
Line 58: please add reference for this statement.
Response 1: Thank you very much for your comment. According to your valuable suggestions, We have added reference for the statements. (Line 59-63, 506-517)
Point 2: Line 83: a very recent publication, Hamel 2023, “Identification of the Tembusu Virus in Mosquitoes inNorthern Thailand ”, described the presence of the TMUV cluster 3 in mosquitoes in North Thailand. This important paper is lacking in this manuscript. Please add this reference.
Response 2: Thank you for your valuable comment. We have added this reference in this manuscript. (Line 87, 553-554)
Point 3: Line 87 to line 97. Please rephrase this part. Here it’s the “introduction part” not the result or discussion part. Introduce the purpose and the question of the study and what you did to answer the question, but without presenting results of the study itself yet. Keep results for results part.
Response 3: Thank you for your important point. We have revised this part according to the comment. (Line 91-94)
Point 4: Line 115-116. This sentence can be deleted as it’s a result and author already addressed this point in the dedicated section.
Response 4: Thank you for your valuable comment. We have deleted this sentence. (Line 115)
Point 5: From Line 117 to Line 192: authors used in all this section the abbreviation “DTMUV” to describe their virus. In all the document, author used “TMUV” to describe their virus. Please, homogenate and used only “TMUV”.
Response 5: Thank you for your careful advice. We have replaced “DTMUV” with “TMUV” in this part. (Line 125-158, 290)
Point 6: Line 136; Please short description of the technic to fully sequence the genome.
Line 137. Describe the alignment: software, method etc..
Response 6: Thank you very much for your comment. We have added related description in this part. (Line 146-154)
Point 7: Line 138: “… fragments and colonies”. Please specify what the term "colonies" refers to?
Response 7: Thank you for your valuable comment. The term "colonies" mean colonies transformed a pCE2 TA/Blunt-Zero vector that contain fragments of PCR products. We have revised this sentence. (Line 151-154)
Point 8: Line 140-144: please add information about what is the model used for phylogenetic analysis. How the author determines this model?
Response 8: Thank you for your careful advice. The neighbor-joining method was used for phylogenetic analysis. (Line 156).The neighbor-joining method is a method of studying DNA to establish kinship and is commonly used in computational biology, bioinformatics, systems biology, evolutionary biology. There are many papers that used this model in phylogenetic analysis:
- 1. Yu, Z.; Ren, H.; Sun, M.; Xie, W.; Sun, S.; Liang, N.; Wang, H.; Ying, X.; Sun, Y.; Wang, Y.;et al. Tembusu virus infection in laying chickens: Evidence for a distinct genetic cluster with significant antigenic variation. Transbound Emerg Dis.2022, 69, (4), e1130-e1141.
- 2. Yan, D.; Li, X.; Wang, Z.; Liu, X.; Dong, X.; Fu, R.; Su, X.; Xu, B.; Teng, Q.; Yuan, C.; et al.The emergence of a disease caused by a mosquito origin Cluster 3.2 Tembusu virus in chickens in China. Vet Microbiol.2022, 272, 109500.
- 3. Fang, Y.; Hang, T.; Yang, L. M.; Xue, J. B.; Fujita, R.; Feng, X. S.; Jiang, T. G.; Zhang, Y.; Li, S. Z.; Zhou, X. N. Long-distance spread of Tembusu virus, and its dispersal in local mosquitoes and domestic poultry in Chongming Island, China. Infect Dis Poverty.2023,12, (1), 52.
Point 9: Line 144: according to the phylogenetic tree, authors used 64 sequences to build it. Please add a table with all sequence used including : Name/accession number/year of isolation/Country.
Response 9: Thank you for your important point. We have add accession number in Figure 2. The phylogenetic analysis displayed the strain name, origin, year of isolation, host and GenBank accession numbers for each virus strain. (Line 291, 295)
Point 10: Line 148-149. Authors said they performed a RT-PCR to screen the presence a set of viruses. Please add a description of the PCR for these viruses including primers sequences used, length of amplicon for each virus, PCR protocol etc… This description can be added to the section 2.3 of the manuscript.
Response 10: Thank you for your valuable comment. We have add a table (1) with all primers used. (Line 134)
Point 11: Line 149. Remove “such as” as only these 4 viruses with the TMUV were tested (according to the figure 1C. Otherwise please, specify all the tested viruses.
Response 11: Thank you very much for your comment. We have removed “such as” according to the comment. (Line 166)
Point 12: Line 204. Remove “unfortunately”.
Response 12: Thank you for your careful advice. We have removed “unfortunately” according to the comment. (Line 227)
Point 13: Line 210-211. Add reference for the antibody if any.
Response 13: Thank you very much for your comment. We have added the information about the antibodies and dyes used. (Line 136-143, 562-564)
Point 14: Figure 1-A. There are only pictures of infected geese organs. Please add pictures of control animal for each organ to compare the physio pathological differences. Like in Fig1-B.
Response 14: Thank you for your careful advice. When we isolated the HQ-22 ,we used only infected geese, not healthy geese. So we could not provide pictures of control animal organs. There are pictures in the published papers that showed only diseased animals organs:
- 1. Yu, Z.; Ren, H.; Sun, M.; Xie, W.; Sun, S.; Liang, N.; Wang, H.; Ying, X.; Sun, Y.; Wang, Y.;et al. Tembusu virus infection in laying chickens: Evidence for a distinct genetic cluster with significant antigenic variation. Transbound Emerg Dis.2022, 69, (4), e1130-e1141.
- 2. Thontiravong, A.; Ninvilai, P.; Tunterak, W.; Nonthabenjawan, N.; Chaiyavong, S.; Angkabkingkaew, K.; Mungkundar, C.; Phuengpho, W.; Oraveerakul, K.; Amonsin, A. Tembusu-Related Flavivirus in Ducks, Thailand. Emerging infectious diseases.2015,21, (12), 2164-7.
- 3. Liu, M.; Chen, S.; Chen, Y.; Liu, C.; Chen, S.; Yin, X.; Li, G.; Zhang, Y. Adapted Tembusu-like virus in chickens and geese in China. Journal of clinical microbiology 2012,50, (8), 2807-9.
Point 15: Line 226: add a table of primers sequences.
Response 15: Thank you for your valuable comment. We have added a table (2) of primers sequences to the section 2.5 of the manuscript. (Line 160)
Point 16: Line 229/230. Which programs were used to assemble fragment and for the alignment (add in section M&M).
Response 16: Thank you for your careful advice. We have added description in this part. (Line 151-154)
Point 17: Line 230: Add the access number in the text.
Response 17: Thank you very much for your comment. We have added the access number(OR909676)for HQ-22. (Line 255)
Point 18: Line 234/236: “Notably … it’s not a result. Move this sentence in discussion section,
Response 18: Thank you for your valuable comment. It has already been expressed in the discussion section, so we have deleted this part. (Line 260)
Point 19: Section 3.2 and Table 1. Author didn’t add in their analyzes the 2 sequences isolated in 2019 and recently described by Hamel et 2023 (Viruses 2023) in North Thailand.
This point is an important point and an important lack in the manuscript as the TMUV cluster 3 isolated in North Thailand represent the latest isolate in this country. Authors also forget to specify that 3 other isolates were reported in Thailand in 1992 and 2002.
Response 19: Thank you for your important point. We have added the 2 sequences isolated in 2019 to the analysis in Table 3 and specified the importance of 3 other isolates were reported in Thailand in 1992 and 2002. (Line 259-262, 290)
Point 20: Please add isolates from Hamel et al in the manuscript and include the sequences to the phylogenetic analysis (add the reference too).
Figure 2. Add the latest strain of TMUV cluster 3 described by Hamel et all 2023 in the phylogenetic tree.
Response 20: Thank you for your valuable comment. We have added the latest strain of TMUV cluster 3 described by Hamel et all 2023 in the phylogenetic tree.(Line 291,553-554)
Point 21: Line 252-261. Please add in this part the latest strain of TMUV cluster 3 described by Hamel et all (Viruses 2023). According to this publication, the amino acid in position 358 of the two sequences is a isoleucine (I) residue. This is the same residue than in strain HQ-22, CTLN and GX2021, it’s so very important to discussion this homology compared to the other Cluster 3 strains presenting a valine (V) on position 358.
Figure 3 Add in the amino acid differences the latest strain of TMUV cluster 3 described by Hamel et all 2023
Response 21: Thank you very much for your comment. We have added the latest strain of TMUV cluster 3 described by Hamel et all 2023 in Figure 3.(Line 278-280,296)
Point 22:Line 296. Please add information about the black arrows in the figure legend.
Response 22: Thank you for your careful advice. We have added the information about the black arrows in the figure legend.(Line 325-328, 366-368)
Point 23: Line 300. Why authors selected ICR mice for this challenge?
Response 23: Thank you very much for your comment. The Institute of Cancer Research (ICR) mice are commonly used experimental animals for constructing pathological models. As is known, mice are good animal models and are used to study the pathogenicity of Flavivirus in mammals :
- 1. Zhu, Y.; Hu, Z.; Lv, X.; Huang, R.; Gu, X.; Zhang, C.; Zhang, M.; Wei, J.; Wu, Q.; Li, J.; et al.A novel Tembusu virus isolated from goslings in China form a new subgenotype 2.1.1. Transbound Emerg Dis.2022, 69, (4), 1782-1793.
- 2. Wang, X.; He, Y.; Guo, J.; Wu, Z.; Merits, A.; Wang, M.; Jia, R.; Zhu, D.; Liu, M.; Zhao, X.; et al.Comparative study of the pathogenicity of the mosquito origin strain and duck origin strain of Tembusu virus in ducklings and three-week-old mice. Virologica Sinica.2023, 38, (5), 827-831.
- 3. Mao, L.; He, Y.; Wu, Z.; Wang, X.; Guo, J.; Zhang, S.; Wang, M.; Jia, R.; Zhu, D.; Liu, M.; et al.Stem-Loop I of the Tembusu Virus 3'-Untranslated Region Is Responsible for Viral Host-Specific Adaptation and the Pathogenicity of the Virus in Mice. Microbiology spectrum.2022, 10, (5), e0244922.
- 4. Shen, T. J.; Chen, C. L.; Jhan, M. K.; Tseng, P. C.; Lin, C. F. CNS Immune Profiling in a Dengue Virus-Infected Immunocompetent Outbred ICR Mice Strain. Frontiers in cellular and infection microbiology2020, 10, 557610.
Point 24: Figure 5 legend. Add the meaning of ic and in in the figure legend.
Response 24: Thank you for your careful advice.We have added the meaning of (i.c.) and (i.n.) in the figure legend. (Line 362, 364)
Point 25: Line 345 “…evolution and mutation.” Add reference (I think ref 20)
Response 25:Thank you for your valuable comment.We have added the reference in this part. (Line 378)
Point 26: Line 355 add reference of strain isolated in Thailand in 2019 (ref Hamel et al 2023)
Line 358 add Hamel et al 2023 for mosquito isolations.
Response 26: Thank you very much for your comment.We have added the description and reference in this part. (Line 388-389, 392)
Point 27: Line 376 to 380. Authors suggest that the substitution A157V have an impact for tissue tropism. But there isn’t any result in their study that support this suggestion. Author use only the strain HQ-22 to infect geese and mice without any comparison with infection with other Cluster 2 and cluster 3 strain. So, it’s not possible for the author to suggest any impact of this amino-acid in the tropism and pathogenesis. Please remove this or rephrase with more with more caution.
Response 27: Thank you for your important point.We have removed this sentence. (Line 409)
Point 28: Line 381 or in other place in discussion: I invite the author to discuss about the amino-acid substitution in position 358 (I358V) with comparison with strain from North Thailand isolated in 2019.
Response 28: Thank you for your careful advice. We have added the discussion about the amino-acid substitution in position 358 (I358V) with comparison with strain from North Thailand isolated in 2019. (Line 410-414)
Point 29: Line 404. Remove “subcutaneous” as this route of injection was not used described in the experiments reported in this manuscript.
Response 29: Thank you for your valuable comment. We have removed “subcutaneous” in this sentence. (Line 431, 439)
Point 30: Line 405-407. It’s not clear. Why authors said the virus evolved and develop mechanism tho breach the blood-brain barrier? Which results support this evolution as former paper describe the existence of airborne route? Here authors did intranasal injection, but there is a possibility to enter in contact and penetrate via olfactive nerves directly. More investigation is needed to support the evolution of infectious mechanism to pass through the BBB. I invite the author to be more cautious in this part of their discussion and to rephrase.
Response 30: Thank you for your important point. Our description is not rigorous enough, We have revised this sentence. (Line 442)
Point 31: Line 407-409. Rephrase because TMUV already leads to severe viral encephalitis in bird.
Response 31: Thank you for your valuable comment. We have revised this senstence. (Line 443)
Point 32: Line 415: “new cluster.. is emerging”. In the manuscript, authors wrote that this cluster 3 is an ancestor of other clusters 1 and 2, so may I know if they consider Cluster 3 as a “new cluster” or if they think this cluster wasn’t reported previously?
Response 32: Thank you for your careful advice. “new cluster” is not rigorous enough, We have replaced “new” with “ less reported”. (Line 450)
Point 33: Please, remove the reference number of every product in Materials and Methods part and homogenate the presentation of reference of product used in this study.
Response 33: Thank you very much for your comment. According to your valuable suggestions, we have revised this part. (Line 126-161, 200-209)
Point 34: Line 53. “.. a solitary open…” not a really good English to describe “ a single open reading frame”
Response 34: Thank you for your important point. We have replaced “solitary” with “single”. (Line 55)
Point 35: Line 63: you can also add the reference (37) here.
Response 35: Thank you very much for your comment. We have added the reference (22) in this sentence. (Line 66)
Point 36: Line 128 and 129: please describe shortly the RNA extraction RT-PCR. Add in a supplementary table set of primers used in this study for the TMUV detection. Describe shortly the fluorescent observation assay.
Response 36: Thank you for your careful advice. We have added the description in two parts. (Line 125-143)
Point 37: Line 147: I think it’s better to write “ were used” instead of “were utilized”.
Response 37: Thank you very much for your comment.We have replaced “were utilized” with “were used”. (Line 164)
Point 38: Line 375 “…cluster3, an ancestral strain, …” used the conditional tense not directly “an ancestral strain”.
Response 38: Thank you for your important point. We have revised this senstence. (Line 409)
Point 39: Line 411. “our study isolated a…”, I think it’s not correct as “the study” didn’t do the isolation itself. Authors did. Please rephrase.
Response 39: Thank you for your valuable comment. We have revised this senstence . (Line 446)
Round 2
Reviewer 2 Report
Comments and Suggestions for Authors
I would like to thank the authors for their answers to my questions and to have corrected points that I raised in my first report.
Thank you very much for the explaination on my question about ICR mice and the references added in the answer.